# FastCPH: Efficient Survival Analysis for Neural Networks

**Xuelin Yang**[*]
Stanford University

**Louis Abraham**
Université Paris 1 Panthéon-Sorbonne

**Sejin Kim**
University of Toronto

**Petr Smirnov**
University of Toronto

**Feng Ruan**
Northwestern University

**Benjamin Haibe-Kains**
University of Toronto

**Robert Tibshirani**
Stanford University

## Abstract

The Cox proportional hazards model is a canonical method in survival analysis for prediction of the life expectancy of a patient given clinical or genetic covariates – it is a linear model in its original form. In recent years, several methods have been proposed to generalize the Cox model to neural networks, but none of these are both numerically correct and computationally efficient. We propose *FastCPH*, a new method that runs in linear time and supports both the standard Breslow and Efron methods for tied events. We also demonstrate the performance of *FastCPH* combined with *LassoNet*, a neural network that provides interpretability through feature sparsity, on survival datasets. The final procedure is efficient, selects useful covariates and outperforms existing CoxPH approaches.

## 1 Introduction

Survival analysis studies the dependency of the time to the occurrence of an event on predictor variables. We usually call the estimated period "duration" and the event of interest "death" or "failure." Censored data, where the endpoint of observation is not a failure, is an important component in this field and requires specialized techniques [2].

The Cox proportional hazards model (CoxPH) is a classic semi-parametric method to handle censored data [1]. It was originally used as a linear regression model, supposing the log risk of failure is a linear combination of the clinical or genetic predictor variables. Its core idea is that the dependency of hazard rate on covariates is time-invariant and multiplicative. The formulation of CoxPH likelihood is explained in Section 2. CoxPH is developed under the assumption of continuous data, but there are often tied event times in real datasets. Several ways have been offered to deal with instances where there are ties, including the exact partial likelihood, the Breslow approximation, and the Efron approximation, the latter two being well-tested in theory and widely used in practice [2, 3, 4].

In classical survival studies using CoxPH, a lot of effort is needed in feature engineering to make the model work well. Over the years, several methods were proposed to generalize it to nonlinear situations, allowing more complex formulation for the log-risk function, yet having mixed results [5, 6, 7]. We review the four most popular implementations of the CoxPH model in neural networks as in Table 1: **DeepSurv [8]** implements a deep learning generalization of the Cox proportional hazards model using Theano and Lasagne. It supports tensor operation, runs in $\mathcal{O}(n)$ by vectorized

---

[*]Correspond to `xyang23@stanford.edu`

[2]We refer censored data as right censoring (the most common type) in this work.

36th Conference on Neural Information Processing Systems (NeurIPS 2022).

Table 1: Comparisons for different Cox PH implementations

|  | Sksurv | PyCox | DeepSurv | PySurvival | *FastCPH* (**Ours**) |
|---|---|---|---|---|---|
| Time complexity | $\mathcal{O}(n)$ | $\mathcal{O}(n)$ | $\mathcal{O}(n)$ | $\mathcal{O}(n^2)$ | $\mathcal{O}(n)$ |
| Deep learning | ✗ | ✓ | ✓ | ✗ | ✓ |
| Handling ties |  |  |  |  |  |
| Tie-awareness | ✓ | ✗ | ✗ | ✓ | ✓ |
| Efron approximation | ✓ | ✗ | ✗ | ✓ | ✓ |
| Breslow approximation | ✓ | ✗ | ✗ | ✗ | ✓ |

cumulative sums over the entire input, but assumes there are no tied events. **Pycox [9]** computes in $\mathcal{O}(n)$ a cumulative sum of all input hazards but not the true risk sets of the CoxPH function. **PySurvival [10]** is a PyTorch compatible implementation based on Deepsurv [8], while adding an $\mathcal{O}(n^2)$ index matrix for Efron's method to handle ties. **Scikit-survival [11]** (sksurv) is package released in 2020 [12] with Brewslow and Efron approximation in $\mathcal{O}(n)$ using an inner for-loop at each distinct event time when going through all events. However, it does not support deep learning. Simply assuming the absence of ties or ignoring all tied cases is statistically inappropriate. Failure to handle ties and oversimplifications may cause unexpected consequences, especially when the behaviors of neural networks possess randomness and the results can be hard to interpret [4, 13]. However, to the best of our knowledge, none of the existing popular survival analysis packages have CoxPH implemented in both an efficient and correct way for neural networks.

Here we present the Fast Cox Proportional Hazard model (*FastCPH*), a computationally efficient and statistically correct method for survival analysis using neural networks. *FastCPH* is a fully vectorized method that runs in $\mathcal{O}(n)$ and yields both standard Breslow and Efron methods for tied events, which overcomes the limitations of existing CoxPH methods. We implement it with PyTorch and it can be easily used for any other machine learning research requiring the CoxPH model, allowing computationally efficient deep learning training for a larger scale of data. We also combine it with *LassoNet* to present *FastCPH-LassoNet*, a simple neural network that provides interpretability through feature sparsity in survival analysis. We evaluate *FastCPH-LassoNet* on multiple survival datasets, and *FastCPH-LassoNet* outperforms existing CoxPH approaches.

## 2 Fast Cox Proportional Hazards Model

We propose *FastCPH* as a method for efficient survival analysis for neural networks. As noted in Table 1, *FastCPH* is the first CoxPH method that is both computationally efficient for neural networks and supports Efron and Breslow approximation in handling tied events. We also provide a step-by-step proof verifying the correctness of our tensor-based implementation.

**Definition 2.1.** The input is given as a feature matrix **x** where each row is a sample, and each column is a feature. Each row is associated with an event time $t_i$ (that can produce ties) and an indicator $\delta_i$ indicating whether the event is censored or not (1 if uncensored). Given a regression model that gives a real number $g(x_i)$ for each sample, the CoxPH likelihood that is maximized in the absence of ties is:

$$L(g) = \prod_{i|\delta_i=1} \frac{\exp(g(\mathbf{x_i}))}{\sum_{j|t_j \geq t_i} \exp(g(\mathbf{x_j}))} \tag{1}$$

The negative log likelihood (loss function) is

$$LL(g) = \sum_{i \in J} \log \left( \sum_{j \in R_i} \exp\left[g(\mathbf{x_j})\right] \right) - g(\mathbf{x_i}) \tag{2}$$

In implementation, assuming the event times $t$ are sorted in decreasing order (which is a $\mathcal{O}(n \log n)$ preprocessing), we can compute all values of $\log \left( \sum_{j \in R_i} \exp\left[g(\mathbf{x_j})\right] \right)$ in $\mathcal{O}(n)$ by using the logcumsumexp function[3] (as implemented in PyTorch [14]):

$$LL(g) = \sum_{i \in J} \texttt{logcumsumexp}(g(\mathbf{x}))_i - g(\mathbf{x_i}) \tag{3}$$

---

[3] $\texttt{logcumsumexp}(g(\mathbf{x}))_i$ is a slight abuse of notation as $\texttt{logcumsumexp}$ is applied on all events, not just those from $J$, then indexed on the elements of $J$.

Table 2: Performance of different CoxPH models on survival datasets (in percentage, 95% CI if indicated). $\diamondsuit$: Models with simple linear architecture (i.e. hidden dimension is 1 and the number of hidden layers is 1). $\clubsuit$: Models with more complex structures, best results from (16,16), (32), (32, 16), (64).

|  | Breast cancer | WHAS500 | Veteran's lung cancer | HNSCC |
|---|---|---|---|---|
| CoxPH Linear | 51.4 | 71.3 | 66.4 | 59.1 |
| CoxNet | 57.0 | 71.4 | 72.6 | 74.3 |
| GlmNet | 60.3 ($\pm$4.67) | 70.1 ($\pm$0.68) | 70.7 ($\pm$1.34) | **74.8** ($\pm$2.18) |
| DeepSurv$^{\diamondsuit}$ | 57.8 ($\pm$1.52) | 70.0 ($\pm$2.05) | 66.1 ($\pm$3.14) | 73.1 ($\pm$1.53) |
| DeepSurv$^{\clubsuit}$ | 68.7 ($\pm$1.20) | 66.1 ($\pm$1.07) | 69.3 ($\pm$0.72) | 61.6 ($\pm$3.05) |
| *FastCPH-LassoNet* $^{\diamondsuit}$ | 67.0 ($\pm$5.39) | 76.6 ($\pm$1.21) | 71.9 ($\pm$1.90) | 70.1 ($\pm$3.96) |
| *FastCPH-LassoNet* $^{\clubsuit}$ | **69.7** ($\pm$5.35) | **76.8** ($\pm$1.40) | **73.0** ($\pm$2.49) | 69.3 ($\pm$4.68) |

**Breslow's method**   When there are ties, the theoretical best solution would assume that the events still happened in some order and sum the formula without ties over all orders. This is not efficient because there are $d_i!$ possible orders for each tie, thus rarely used in applications. Breslow approximation assumes that all $d_i$ elements were selected from the same risk set. Thus, the above formula stays unchanged. The implementation simply indexes the `logcumsumexp` terms so that events with the same failure time have the same denominator.

**Efron's method**   observes that the denominator is too big in Breslow's approximation, as when multiples elements are selected from the same risk set, that risk set gets smaller.

**Definition 2.2.** Efron's approximation results in the following likelihood:

$$L(g) = \prod_{i \in J'} \frac{\prod_{j \in D_i} \exp(g(\mathbf{x_i}))}{\prod_{k=0}^{d_i-1} \left( \sum_{j \in R_i} \exp(g(\mathbf{x_j})) - \frac{k}{d_i} \sum_{j \in D_i} \exp(g(\mathbf{x_j})) \right)} \tag{4}$$

where $J'$ is a subset of $J$ with unique event times. The idea is to discount the denominator over the whole risk set $\sum_{j \in R_i} \exp(g(\mathbf{x_j}))$ by the average risk over $D_i$: $\frac{1}{d_i} \sum_{j \in D_i} \exp(g(\mathbf{x_j}))$. $k$ indexes the set $D_i$.

Compared with Equation 2, the numerator did not change (it is still the product over all uncensored events). The denominator has three terms:

- $\sum_{j \in R_i} \exp(g(\mathbf{x_j}))$ can be computed for all $j$ in linear time as before.
- $\frac{k}{d_i}$ is trivial to compute for all elements of $J$ (which is the union of all $D_i$ for $i \in J'$).
- $\sum_{j \in D_i} \exp(g(\mathbf{x_i}))$ can also be computed in linear time with a vectorized scatter operation.

Finally, those terms are easy to combine in linear time with vectorized operations. All computations are realized in the log space to avoid numerical errors, using tricks similar to those used to implement the log-sum-exp operation. To summarize, we provide *FastCPH* as an efficient, vectorized and linear-time function implemented in PyTorch that can be conveniently used by any neural network.

***FastCPH-LassoNet***   is proposed as a survival prediction method with feature sparsity. We use *LassoNet* as the backbone [15], which is a neural network that achieves feature sparsity using a LASSO-style regularization [16]. Originally, it was applied with mean squared error and cross-entropy losses for regression and classification problems. We apply *FastCPH* as the loss function for survival analysis. The implementation detail is in Section A.2 in the supplementary materials.

## 3   Experiments

In the experiments, we want to evaluate the following three aspects of *FastCPH*: Is the implementation of *FastCPH* computationally efficient for neural networks? Can *FastCPH-LassoNet* obtain

Table 3: Hyperparameters and run time of *FastCPH-LassoNet* (linear structure, 95% CI if indicated). Run time is in seconds for per run per CPU. Training performed on 2.8 GHz Quad-Core Intel Core i7 with 16 GB memory.

|  | Breast cancer | WHAS500 | Veteran's lung cancer | HNSCC |
| --- | --- | --- | --- | --- |
| # selected features | 24.6 (±2.95) | 14.0 (±0.00) | 10.6 (±0.78) | 11.0 (±3.95) |
| # total features | 80 | 14 | 11 | 107 |
| run time | 261s | 225s | 201s | 283s |

feature sparsity along the regularization path? Does *FastCPH-LassoNet* have promising performance compared to other CoxPH-based models on real-world survival datasets?

## 3.1 Runtime analysis

To analyze the computational efficiency of *FastCPH*, we compare its runtime with 3 other popular vectorized implementations including PyCox, DeepSurv, PySurvival. PySurvival use risk / fail matrices to compute Efron method. PyCox and DeepSurv are for deep learning purposes and do not support tie approximations.

Among these baselines, *FastCPH* is the most computationally efficient CoxPH implementation that supports Breslow and Efron approximations. Specifically, as shown in Fig 1, the curve of *FastCPH* with Breslow method is very close to the ones of PyCox and DeepSurv, which don't have tie awareness. As the size of data increases, the difference among *FastCPH* with Breslow, PyCox, and DeepSurv are getting smaller. For the two baselines both using Efron approximation, *FastCPH* with Efron method is significantly faster than PySurvival. The gap between *FastCPH* with Efron and Breslow can be caused by the computation of the weighted terms in denominator. The experimental results align with our claims on the linearity of *FastCPH*'s runtime.

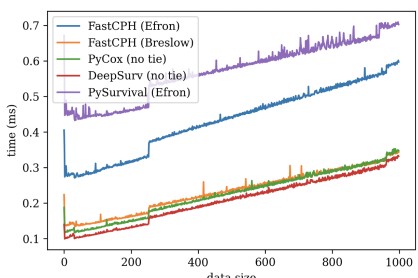

Figure 1: Runtime comparisons between different CoxPH implementations. The x-axis is the size of data, and the y-axis is the time for one-time CoxPH calculation in milliseconds

## 3.2 Evaluating *FastCPH-LassoNet* on real world's datasets

We compare *FastCPH-LassoNet* with other CoxPH-based methods on 4 datasets: breast cancer [17], WHAS500 [18], veteran's lung cancer [19], and HNSCC [20]. We use CoxPH linear model [1], CoxNet [21], GlmNet [22] and DeepSurv [8] as baselines. The first three are classical CoxPH-based models with different regularizations. DeepSurv is considered the most advanced deep learning CoxPH-based method [8, 23]. We use Harrell's concordance index (C-index) as a metric. We also provide a $\mathcal{O}(n \log n)$ implementation of the C-index using ordered data structures.

As in Table 2, *FastCPH-LassoNet* is the best CoxPH-based method on 3 out of 4 survival datasets we use, showing its discrimination ability to provide reliable ranking of survival times based on risk scores. Notice that for the breast cancer and WHAS500 datasets, *FastCPH-LassoNet* yields a significantly better C-index than DeepSurv and other existing methods in Table 2. Specifically, comparing *FastCPH-LassoNet* to DeepSurv, *FastCPH-LassoNet* performs better in 4 datasets under the same architecture. We supplement further comparisons between *FastCPH-LassoNet* and DeepSurv in Section A.2.

Moreover, *FastCPH-LassoNet* is able to attain an effective recovery of signals given a subset of features. As shown in Table 2 and 3, the model gives excellent performance with sparsity in covariates. However, when the number of total features is small, the model may not be able to obtain sparsity over covariates, as shown by its result of WHAS500. The sparsity demonstrated in the experiments implies its potential on large-scale, more complicated real world datasets.

## 4 Discussion

In this paper, we have proposed *FastCPH*, an efficient CoxPH method for survival analysis in neural networks that follows the exact formula of well-tested methods to handle tied events. *FastCPH* is an efficient and numerically correct solution for neural networks in survival analysis. It can be quickly adapted to other deep learning methods and used in more real-world scenarios with censored data such as [24, 25, 26]. We have shown *FastCPH-LassoNet* outperformed other CoxPH-based methods in various survival datasets. More study can be done to provide applications of *FastCPH* as an objective function in more complex neural network architectures. It will be interesting to see the effect of tied events on the behavior of neural networks. In addition, it is worth pointing out the importance of following the exact formulae of classic statistical methods in implementation and avoiding oversimplifications in the machine learning community.

## A Appendix

This supplementary document is organized as the following. Firstly, we discuss the related work on CoxPH models, deep learning in survival analysis, and tie handling. Next, we discuss implementation details on the metric, settings, and baselines. We then provide additional information on datasets we used in the experiments, including an illustration of pipeline, covariate breakdown, and the correlation matrix of HNSCC dataset. Lastly, we provide additional experiment comparing *FastCPH-LassoNet* with DeepSurv in NN context. We also supplement the code for *FastCPH* on `https://github.com/lasso-net/lassonet`.

### A.1 Related Work

**CoxPH models**   A major advantage of CoxPH models over methods like Kaplan-Meier curves and the log-rank test is their ability to work easily with quantitative predictor variables and ability to generalize patterns from censored data [27]. Therefore, they are particularly suitable for survival analysis and predictions and are applied extensively in the biomedical field including in analyzing gene expressions and the likelihood of various diseases including liver diseases, coronary heart disease, diabetes, etc [28, 29, 30, 31]. Beyond that, CoxPH models also have a variety of applications. When compared with the results of the multiple discriminant analysis methods, the CoxPH model gives lower type I errors [24].

**Deep learning in survival analysis**   With the rise of deep learning applications in many scientific fields, some studies have tried to combine CoxPH functions with deep neural networks for better time-to-event predictions for larger datasets [8, 23, 32]. The inclusion of a neural network can simplify the *a priori* covariates selection and make the model adaptively learn them while preserving the effectiveness of CoxPH functions in survival analysis. Our work focuses on the deep learning method to model the survival hazards using CoxPH.

**Handling ties**   When handling ties, the most commonly used approach is the Breslow method which simply uses the number of tied events as the exponent in the denominator of the relative risk [2]. The Efron method is thought to produce superior outcomes, yet the formulation is more complicated to implement efficiently [3]. The difference with the Breslow approach is minor when the number of ties is not large [33]. Our method supports both Efron's and Breslow's method.

### A.2 Implementation Details

**Metric**   Harrell's concordance index (C-index) is a generalization of Area under ROC curve (AUC) regarding censored data, reflecting the accuracy of pairwise orders of the risk function as the output of the model [34, 35, 36]. For input $\mathbf{x}_i$, duration $t_i$ and events $\delta_i$ (1 if uncensored, 0 otherwise), the C-index is computed as:

$$C = \frac{\sum_{i,j:t_j<t_i} 1_{g(\mathbf{x}_j)>g(\mathbf{x}_i)}\delta_j}{\sum_{i,j:t_j<t_i} \delta_j}. \tag{5}$$

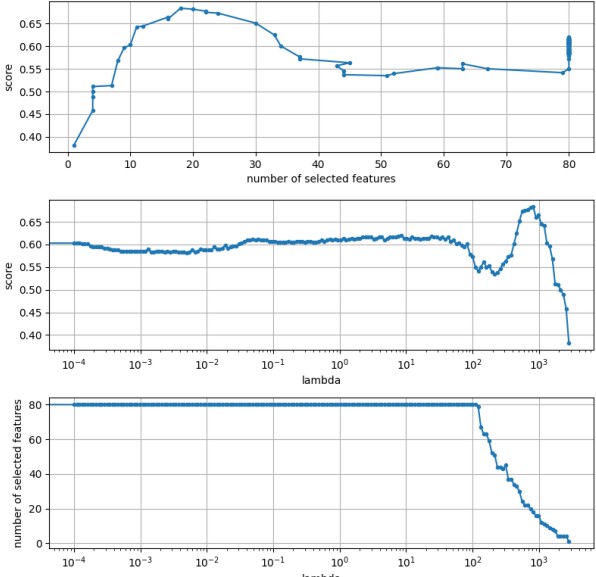

Figure 2: Demonstration of the training behavior of *FastCPH-LassoNet* on the breast cancer dataset. The y-axis of the first two rows represents the test score, which is calculated as the C-index. The model starts from a given $\lambda$. At each iteration, $\lambda$ increases at a fixed geometric rate, the model is retrained based on the model selected at the last $\lambda$. At each $\lambda$, the iterated model can select a subset of features in the input, that is supposed to decrease. The first figure is the number of selected features versus the test C-index. The second figure represents the change in test score when $\lambda$ increases during training. The third figure is the number of features selected when $\lambda$ increases during training. We can see that the test score reaches a peak when $\lambda$ is between $10^2$ and $10^3$. The test C-index fluctuates when the number of selected features decreases, and it reaches the global optimal when the number of features is 19.

**FastCPH-LassoNet**  is *LassoNet* combined with *FastCPH* as the objective function. Like LASSO, *LassoNet* penalizes the $L^1$ norm of coefficients applied to features. The Lagrange multiplier associated with that penalty is noted $\lambda$. The model is trained with increasing values of $\lambda$, on a dense-to-sparse path where the values of $\lambda$ follow a geometric scale. The starting value of $\lambda$ is a hyperparameter that should be carefully selected: if it is too small, the model will train on a lot of useless configurations; if it is too large, the optimizer will ignore features too fast. Another hyperparameter is $M$, the hierarchy coefficient that balances the linear and non-linear parts of the model. Fig 2 gives an example of the training curve of *FastCPH-LassoNet* on the breast cancer dataset. We can see *LassoNet* optimized properly with *FastCPH* as the loss function.

**Runtime Analysis**  We assume all events are uncensored and the data is sorted by duration. We first randomly generate data from size 1 to $10^3$. We note down the runtime as calculating the negative log likelihood once and report the mean of 5 random trials.

**Baseline implementation**  The implementations of the Cox linear model and CoxNet are from Scikit-survival [11]. For the CoxPH linear model, we set $\alpha = 10^{-6}$ as the regularization parameter in the ridge regression penalty. CoxNet is the CoxPH model with an elastic net penalty. We use cross-validation for choosing the best $\alpha$ of the regularization path from $10^{-1}$ to $10^{-5}$. For GlmNet, we use the R built-in cross-validation `cv.glmnet` with the Breslow method to select the model for testing. The number of folds in cross-validation (if used) is 5 for breast cancer and veteran's lung cancer dataset and 10 for WHAS500 and HNSCC dataset. For *FastCPH-LassoNet* and DeepSurv, we fix the number of hidden dimensions to 1 and the number of hidden layers to 1. They share the same architecture and setting (ReLU, Adam, lr=$10^{-3}$). For *FastCPH-LassoNet*, we set $M = 10$ and start

at $\lambda = 10^{-6}$. The `prox` method of *LassoNet* is called on the dense model following on a geometric path until the model becomes sparse. That value is divided by 10 to give `lambda_start`. 5-fold cross validation is used to select the best $\lambda$ value from multiple runs. We use Efron's method to break ties. We use stratified sampling w.r.t uncensored/censored events to split the training set (80%) and test set (20%) for each of the datasets. For models with randomness in training, we run 5 trails for each set of hyperparameters and obtain the average performance.

## A.3 Datasets statistics

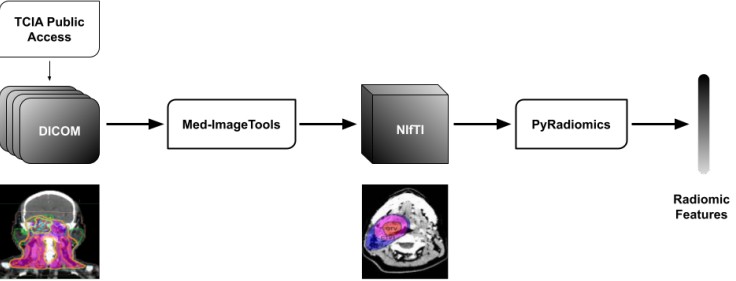

Figure 3: The pipeline of obtaining HNSCC features for survival analysis. HNSCC dataset was curated by the University of Texas MD Anderson Cancer Center, approved by the institutional review board, and written informed consent was obtained from all study participants. This dataset is available publicly upon submission of a limited data access agreement to safeguard patient privacy. While the dataset has been thoroughly de-identified in accordance with HIPAA, it is theoretically possible to reconstruct the face, head, or body using volumetric images. For our analysis, we ensured the data was processed on a HIPAA-compliant encrypted server and the radiomics features extracted for our analysis do not contain any identifiable information or offensive content.

We use the following four datasets in our experiments:

- **Breast cancer dataset [17]:** This dataset comes from experiments set up to validate a certain gene signature in primary breast tumors. It contains data points from 198 patients, with 80 features each. The endpoint of this dataset is distant metastases. Of all patients, 51 of them (25.8%) exhibited the symptom.

- **WHAS500 dataset [18]:** The Worcester Heart Attack Study dataset is an observational dataset set up to track trends in acute myocardial infarction and out-of-hospital coronary heart disease deaths in Worcester, Massachusetts. The endpoint of this dataset is death. Out of 500 patients in the dataset with 14 features each, the endpoint occurred for 215 patients (43.0%).

- **Veteran's lung cancer dataset [19]:** This dataset comes from a lung cancer trial by the Department of Veterans Affairs. The endpoint of this dataset is death. Out of 137 patients in the dataset with 6 features each, the endpoint occurred for 128 patients (93.4%).

- **HNSCC [20]:** This dataset is composed of 451 head and neck squamous cell carcinoma (HNSCC) patients treated with curative-intent intensity modulated radiotherapy (IMRT). This dataset was previously used to predict local recurrence and HPV status [37]. Survival analysis is done in data exploration, but nothing predictive. We include it as a showcase of our method. The endpoint used for our analysis is death. This is a challenging prediction dataset when adpating it for survival analysis. Fig 4 is a visualization of the correlation matrix of clinical covariates in HNSCC dataset generated by PySurvival [10]. As we can see in the figure, many of the covariates are heavily correlated with each other, posing a need of

Table 4: Event and clinical variable distribution of HNSCC.

| # of patients | 451 |
|---|---|
| **Outcome** | |
| Alive / Censored | 395 (88%) |
| Death | 56 (12%) |
| **Sex** | |
| Male | 387 (86%) |
| Female | 64 (14%) |
| **Disease subsite** | |
| Base of Tongue | 238 *(53%)* |
| Tonsil | 174 *(39%)* |
| Glossopharyngeal sulcus | 10 *(2%)* |
| Other | 29 *(6%)* |
| **HPV Status** | |
| Positive | 232 *(51%)* |
| Negative / Unknown | 219 *(49%)* |
| **Stage** | |
| I | 3 *(1%)* |
| II | 14 *(3%)* |
| III | 63 *(14%)* |
| IV | 371 *(83%)* |
| **Tumor Laterality** | |
| R | 222 *(49%)* |
| L | 215 *(48%)* |
| Other | 14 *(3%)* |

Table 5: Statistics on tied events in different datasets

| | Breast cancer | WHAS500 | Veteran's lung cancer | HNSCC |
|---|---|---|---|---|
| # total observations | 198 | 500 | 137 | 451 |
| # total tied events | 6 | 178 | 64 | 232 |
| # uncensored events | 51 | 215 | 128 | 56 |
| # uncensored tied events | 0 | 80 | 55 | 4 |

selecting useful features for constructing an efficient solution. According to a single value decomposition computation, the matrix is rank deficient. Despite that, *FastCPH-LassoNet* can successfully select a subset of covariates (11.04 out of 107) and attain a good and stable performance as shown in Table 2.

**Ties** are common in the datasets we use, listed as in Table 5. Breslow and Efron methods give the same log likelihood when no ties are present in the dataset. Our method in Table 2 using Breslow approximation is capable of handling datasets with and without ties.

**Data preprocessing** For the breast cancer dataset, WHAS500, and veteran's lung cancer dataset, we retrieve the data from Scikit-survival package [38] and obtain one-hot encodings to quantify entries such as treatment received, cell types, prior therapy, etc. The number of final entries is shown in the last row of Table 3. The HNSCC dataset is publicly available via the Cancer Imaging Archive with TCIA Limited Access License [20]. The DICOM imaging data is processed using the Med-ImageTools pipeline [39] to extract the computed tomography (CT) images and gross tumor volume (GTV) segmentation masks with uniform voxel spacing for consistent feature extraction. These images and masks are processed into the NIfTI file format, which is a common standard for 3D medical images, and compatible with PyRadiomics. The processed image and GTV masks into PyRadiomics to extract shape, texture, and statistics features [40].

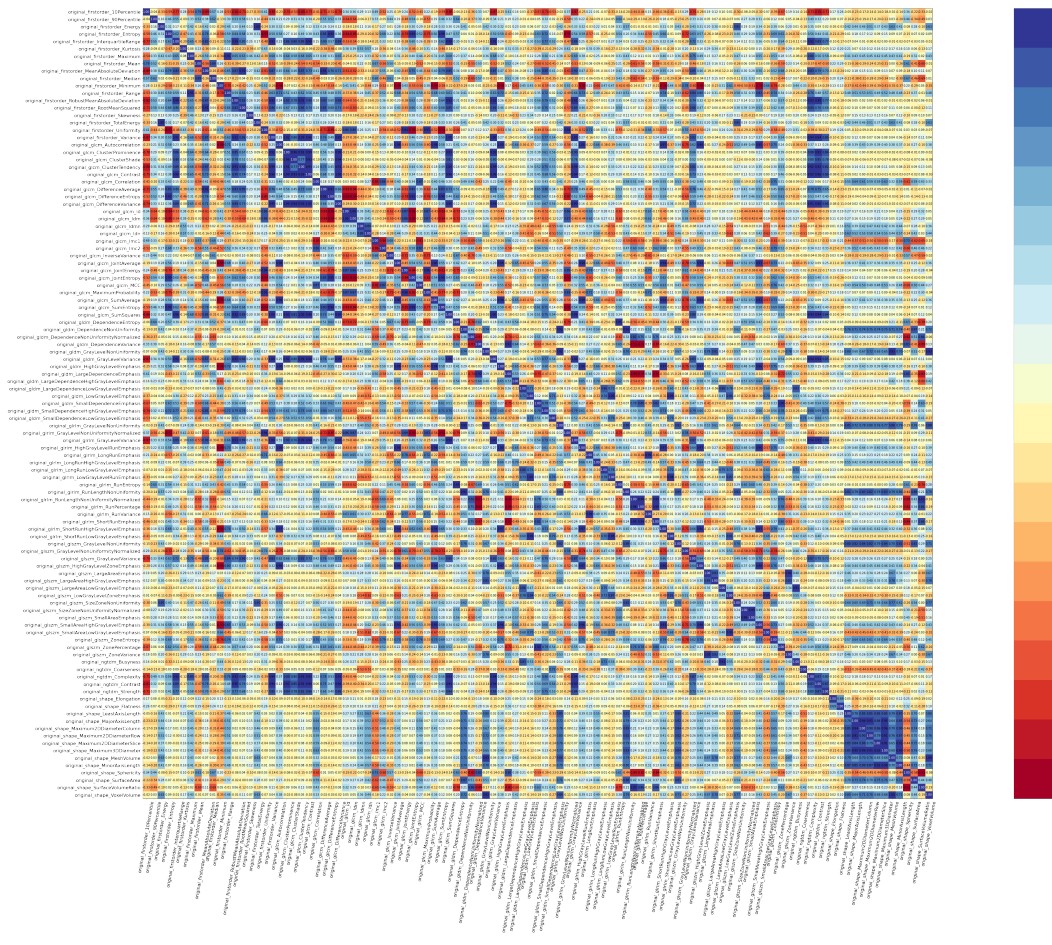

Figure 4: Correlation matrix of covariates in HNSCC. The red color at the bottom of the spectrum indicates −1 and blue color at the top means 1.

Table 6: Results of *FastCPH-LassoNet* and DeepSurv with more complex architectures (in percentage, 95% CI)

|  | Breast cancer | WHAS500 | Veteran's lung cancer | HNSCC |
|---|---|---|---|---|
| (16, 16) | | | | |
| *FastCPH-LassoNet* | **69.7** (±5.35) | 76.6 (±1.35) | **73.0** (±2.49) | 63.7 (±5.89) |
| DeepSurv | 68.2 (±2.47) | 64.5 (±1.06) | 66.3 (±1.47) | 61.6 (±3.05) |
| (32) | | | | |
| *FastCPH-LassoNet* | 67.4 (±4.90) | 76.8 (±1.29) | 72.6 (±2.12) | **69.3** (±4.68) |
| DeepSurv | 66.6 (±1.27) | 65.8 (±0.68) | 69.3 (±0.72) | 58.9 (±2.56) |
| (32, 16) | | | | |
| *FastCPH-LassoNet* | 69.0 (±3.47) | 75.4 (±2.49) | 71.9 (±3.23) | 65.3 (±5.38) |
| DeepSurv | 68.7 (±1.20) | 66.1 (±1.07) | 66.2 (±1.46) | 57.1 (±2.39) |
| (64) | | | | |
| *FastCPH-LassoNet* | 68.6 (±5.11) | **76.8** (±1.40) | 72.9 (±2.50) | 69.0 (±4.22) |
| DeepSurv | 67.0 (±1.24) | 64.8 (±0.63) | 67.3 (±1.13) | 58.5 (±2.00) |

### A.4 *FastCPH-LassoNet* with more complex architectures

In the context of NN methods, we further analyze the performance of *FastCPH-LassoNet* using more complex architectures. We use DeepSurv as the baseline because it is commonly recognized as the most advanced CoxPH-based deep learning method. We let *FastCPH-LassoNet* and DeepSurv share the same architecture (as indicated in parentheses in Table 6) and setting (ReLU, Adam, $\mathtt{lr}=10^{-3}$). For both methods, we run 15 trails to give 95% CI.

As shown in Table 6, *FastCPH-LassoNet* outperforms Deepsurv with the same architecture on all datasets in our experiments. It is a more robust deep learning method with promising results in survival analysis.

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
