# OpenReview forum: "FastCPH: Efficient Survival Analysis for Neural Networks"
_NeurIPS.cc/2022/Workshop/HITY — HITY Workshop NeurIPS 2022_

### Official Review · Reviewer_imci · 2022-10-05
**This paper proposes FastCPH, a CosPH method for survival analysis in neural networks**

**Rating:** 1
**Confidence:** 2

**Review:**

The authors introduce FastCPH as a novel CoxPH implementation. It runs in O(n), enables deep learning while including tie-awareness as well as efron/breslow approximations. It is the first method that does this, and I hence vote 'accept'.

---

### Official Review · Reviewer_bSRe · 2022-10-12

**Rating:** 1
**Confidence:** 2

**Review:**

This paper proposes an efficient method for survival analysis using neural networks. I'm not familiar with the topic, but the paper is well-written and initial results seem to look good.

---

### Decision · Program_Chairs · 2022-10-20

Accept